# Lived Experience of Dementia in the New Zealand Indian Community: A Qualitative Study with Family Care Givers and People Living with Dementia

**DOI:** 10.3390/ijerph19031432

**Published:** 2022-01-27

**Authors:** Rita V. Krishnamurthi, Ekta Singh Dahiya, Reshmi Bala, Gary Cheung, Susan Yates, Sarah Cullum

**Affiliations:** 1National Institute for Stroke and Applied Neurosciences, Auckland University of Technology, Auckland 0627, New Zealand; ekta.dahiya@aut.ac.nz; 2Department of Psychological Medicine, University of Auckland, Auckland 1142, New Zealand; srai816@aucklanduni.ac.nz (R.B.); g.cheung@auckland.ac.nz (G.C.); susan.yates@auckland.ac.nz (S.Y.); sarah.cullum@auckland.ac.nz (S.C.)

**Keywords:** caregivers, dementia, Indian community, New Zealand, qualitative

## Abstract

Currently, there are estimated to be 70,000 people living with dementia in Aotearoa, New Zealand (NZ). This figure is projected to more than double by 2040, but due to the more rapid growth of older age groups in non-European populations, prevalence will at least triple amongst the NZ Indian population. The impact of dementia in the NZ Indian community is currently unknown. The aim of this study was to explore the lived experiences of NZ Indians living with dementia and their caregivers. Ten caregivers (age range: 41–81) and five people living with mild dementia (age range: 65–77) were recruited from a hospital memory service and two not-for-profit community organisations in Auckland, Aotearoa, NZ. Semi-structured interviews were conducted by bilingual/bicultural researchers and transcribed for thematic analysis in the original languages. Dementia was predominantly thought of as being part of normal ageing. Getting a timely diagnosis was reported as difficult, with long waiting times. Cultural practices and religion played a large part in how both the diagnosis and ongoing care were managed. Caregivers expressed concerns about societal stigma and about managing their own health issues, but the majority also expressed a sense of duty in caring for their loved ones. Services were generally well-received, but gaps were identified in the provision of culturally appropriate services. Future health services should prioritise a timely diagnosis, and dementia care services should consider specific cultural needs to maximise uptake and benefit for Indian families living with dementia.

## 1. Introduction

Dementia is a rapidly growing health issue worldwide, with over 50 million people living with the condition [1]. Dementia has a significant health, psychological, and economic impact on the affected individuals and their family caregivers. People living with dementia are likely to become partially or fully dependent on others for support and care, while caregivers often suffer from caregiving burden resulting in health issues and economic losses themselves [2].

In Aotearoa, New Zealand (NZ), people of Indian origin comprise 4.7 percent of the total population, with about 2.5 percent born in India and 1 percent people of Indian ethnicity originating from the Fiji Islands [3]. The majority of Indians (approx. 75%) were born overseas. Indians in NZ are proportionally much more likely to be younger and have more post-high school qualifications than the national average. Approximately 60% are employed full-time; however, their mean income is lower than the national average. In terms of origin, the majority of Indians in NZ hail from India, while a smaller but substantial proportion are from Fiji (described as Fijian Indian). Fijian-Indians are an ethnic group from the Pacific Islands of Fiji. (Ref Stats NZ https://www.stats.govt.nz/tools/2018-census-ethnic-group-summaries/fijian-indian, accessed on 21 November 2021).

This study was conducted in Auckland, the largest city in Aotearoa, NZ, with the highest proportion of Indians in NZ at 10% of the population [4]. These numbers are expected to quadruple in the next 20 years due to longer life expectancy and migration. Currently, there are estimated to be 70,000 people living with dementia in Aotearoa, NZ. The prevalence of dementia in the whole NZ population is projected to more than double by 2040, but the greatest increase is expected in the Asian population due to more rapid demographic ageing [5].

No data have been published about the extent and impact of dementia in the Aotearoa, NZ Indian community. Anecdotal evidence (in Auckland) suggests increasing numbers of referrals of people from Indian and Fijian-Indian backgrounds to local memory services, but there is a lack of understanding of their specific experiences and needs. For these reasons, it is important to gain a better insight into this population’s understanding of dementia and their lived experiences, which could inform the design and implementation of culturally appropriate dementia care and support services as well as prevention strategies.

The aim of our study was to explore how dementia is perceived, experienced, and managed from the perspective of Indian people in Aotearoa, NZ with dementia and their family caregivers.

## 2. Materials and Methods

### 2.1. Participants and Settings

Participants were recruited through three existing services. These included (1) the memory service at Middlemore Hospital (Counties Manukau Health), a large public hospital in South Auckland, (2) a not-for-profit community dementia care organisation (Dementia Auckland) that offers services and support to those at various stages of the journey with dementia, and (3) another non-profit organisation (Shanti Niwas Charitable Trust) which provides support services and community programmes to senior citizens of Indian origin in the Auckland region.

Potential participants from the memory service were initially contacted by a member of the clinical team who asked for consent to pass on contact details to the research team. Those who agreed to be contacted were approached by a member of the research team who gave further details before asking for consent. Participants from the not-for-profit organisations were informed about the study by their community coordinator, and those who agreed to be contacted were approached by a member of the research team. We aimed to be as inclusive as possible but excluded people who were residing in care homes or who could not be interviewed in English, Hindi, or Fijian Hindi. People with more severe dementia were not included in the study; however, their caregivers (regarded as the person most involved in the person living with dementia’s day-to-day life) were interviewed. A voucher worth NZ$50 was provided to the participant at the end of the interview to acknowledge their input in the study.

### 2.2. Semi-Structured Interviews

The interviews were conducted by a team of five lay interviewers (both male and female) who were bilingual and bicultural (Indian or Fijian Indian) with a mix of health research and public sector professional backgrounds. Interviewers attended five training workshops organised by the research team prior to study commencement. Training workshops included an introduction to qualitative research, semi-structured interviews, thematic analysis, the process of obtaining informed consent, discussion on potential questions, tips on managing difficult interviews, and the process of transcribing interviews. The topic guide (Appendix A) guided interviewers’ explorations of the understanding of dementia, experience of the diagnostic process, adjustments made to lifestyle, available support services, and any concerns or recommendations. Caregivers were also asked about their caregiving experiences, any concerns or issues they were facing, and their satisfaction with the level of formal and informal support received.

Interviews were conducted by one or two interviewers at the participant’s house in their preferred language, including Hindi, Fiji Hindi (a dialect of Hindi spoken by people of Indian origin born in Fiji), and English. The researchers introduced themselves to the participants and provided a brief background about themselves prior to the interview, including the reasons for their own interest in the area. Demographic information from the participant and their caregiver was collected. Face-to-face interviews were conducted between July 2018 and January 2019. A peer-review workshop was held after the first two interviews to discuss interviewer experiences and concerns. Each interview was audio-recorded and transcribed verbatim by the interviewers. The transcripts were uploaded into a Computer Assisted Qualitative Data Analysis Software, NVivo (Version 12, QSR International, Melbourne, Australia), to organise the data for coding [6].

### 2.3. Data Analysis

Interview transcripts were analysed by two researchers (RK and ESD) who are fluent in Hindi, Fiji Hindi, and English using the principles of thematic analysis [7]. Both RK and ESD brought their understanding of Indian culture and clinical knowledge in analysing and interpreting the data. The six phases as described by Braun and Clarke formed the basis of analysis: familiarization with the data, generating codes with deductive orientation, constructing themes, revising and defining themes, and producing the reports [7]. The transcripts were coded by both researchers independently and reviewed for recurrent themes during research team meetings.

All participants were assigned a participant identification (ID) number to maintain confidentiality while analysing the data. The relevant quotes used for the analysis were translated to English for the purpose of writing reports and referred to only their ID. A summary table was drafted based on the topic guide, and relevant codes along with specific quotes were added to it. Themes and subthemes were generated from recurring codes and refined further.

### 2.4. Ethics

Ethics approval was gained from the National Health and Disability Ethics Committee (reference number 17/CEM/126/AM01).

## 3. Results

Sixteen participants were initially approached for participation in the study. Of these, a total of fifteen participants with an age range of 41 to 81 years agreed to be interviewed. Five of the interviews were with dyads including people with mild to moderate dementia (who had capacity to consent to participate in the study) and their caregivers, and five interviews were conducted with caregivers only (of people with more severe dementia who were unable to participate in the study). This was sufficient to reach data saturation.

In all but one interview, only the interviewers and interviewees were present, but in one interview, other family members were present (daughter and grandchildren); however, they did not participate. The average interview duration was 35 minutes but varied from 17 to 47 minutes.

The demographic details of all participants are shown in Table 1. Table 2 lists the ethnicity and relationship to the person with dementia for each recruited family caregiver.

### 3.1. Thematic Analyses

The findings from this study are summarised in Figure 1 by linking the main topic themes and subthemes related to caregivers and people living with dementias. There was an overlap of some of the sub-themes, particularly between the caregiver experiences and those of people living with dementia.

We have not included the participant IDs with quotations in the report as it would have been easy to re-identify participants from the sociodemographic tables due to the small population living in the community.

#### 3.1.1. Theme 1: Understanding of Dementia

Subtheme 1.1: Dementia is a normal part of ageing

Participants were asked about their understanding of dementia. Most participants were not able to define the term “dementia” because there is no equivalent common spoken Hindi word for it. However, when the term was further described by the interviewers as “the illness of forgetfulness or being confused,” most participants were able to recognise the condition that was being described. Most participants described dementia as a natural consequence of ageing, and as weakness of the mind and body.


*…this is normal…because he is getting older, he forgets. (C)*


Subtheme 1.2. Dementia may be caused by external stressors or ‘karma’

Other people associated the onset of dementia to stressful life events such as the loss of a job, overall stress, loneliness (particularly associated with migrating to another country), and a lack of physical and mental activities.


*I see one person…he lost his job, he got dementia. (C)*



*There is loneliness here, without our own people. (C)*


Dementia was also described as being due to ‘karma,’ a common perception of the causes of one’s destiny in the Indian culture, particularly amongst Hindus. The concept of karma says that life’s destiny, or fate (‘rekha’), is determined by a person’s past and present deeds. However, the concept of karma brought with it a feeling of acceptance, as karma is not in a person’s direct control. Participants and their families appeared to accept dementia as part of their destiny. However, attributing cognitive impairment to karma might also impact whether people access services in a timely fashion and receive the support they need.


*You are (un)lucky to get Alzheimer or unlucky to get dementia (laughs) (hmm hmm). That’s your, know your karma, it a rekha aur hmm depending in the lifestyle also you get dementia or not get dementia you know. (C)*


#### 3.1.2. Theme 2: Diagnostic Process

Subtheme 2.1: Difficulty in getting a dementia diagnosis

Most people living with dementia and/or their caregiver did not think they got a clear message about their diagnosis, much less about their prognosis. Investigations such as neuroimaging and laboratory tests were not clearly explained. Medications were taken without any real understanding of what they were for. Many found the experience of getting a diagnosis challenging, either because their doctor did not appear to take their concern seriously or because they could not get a timely appointment with a specialist. Caregivers were often told that their family member had an “age related condition” which was frustrating and may explain why many of the participants thought dementia was a normal part of ageing (Theme 1).


*She was very aggressive and… so I took her to the GP and then the GP said ummm… “she’s fine and old age, nothing to…” you know that, but I told to GP “no she’s definitely got some kind of an issue because you know, uhh, cos I know, my mum”. (C)*


Others who originated from India found it difficult to access timely services in Aotearoa, NZ and ended up travelling to India for their health assessment. They reported extensive delays in getting appointments in the public health system and little or no follow-up after their initial appointment with a doctor.


*I had my scan in India, that’s what the doctors here used to diagnose me…doctor said its good you got the scan done. (P)*


Subtheme 2.2: Lack of information on services

The general perception observed was that people with dementia and their caregivers were not provided with sufficient information to process the diagnosis of dementia. Also, there was little or no follow-up made to monitor their condition. For example:


*GP referred to someone else, a girl came from Middlemore about 5–6 months before and asked about memory loss problem…like you people are asking about…and did not contact again…. (P)*


#### 3.1.3. Theme 3: Impact on Personal and Social Life (of Person Living with Dementia)

Subtheme 3.1: Losing independence

Participants with dementia talked about how dementia impacted their independence and reduced their ability to participate in everyday activities such as cooking, gardening, and going out for walks. Both participants living with dementia and their caregivers described these activities as being potentially unsafe, for example, the risk of leaving the stove on, wearing the wrong shoes, or falling over while walking. These changes had developed gradually with time and were ongoing, adding to the stress experienced by both the person living with dementia and the caregiver.


*Cooking… has got stopped altogether for her. We used to prepare meals together earlier. She used to cook alone but children stopped it. Doctor also advised not to go near stove. (C)*



*She prepared dinner today. She manages if I am at home…had to remind her every time to have dinner. (C)*



*Like I use to do a job…so this is my car (pointing out to his car)…I use to drive this…yes…so I use to go and take interviews from people…hmmm…on health issues…hmm…so I use to like that so much that I didn’t want to sit, …yes…when I got stroke, my driving also stopped. (P)*


Subtheme 3.2: Conflict and frustration between the person with dementia and caregiver

The increasing inability to perform such activities was a source of ongoing conflict between the person with dementia and the caregiver.


*Like the routine house chores, she feels that she does that…hmm…but body…her body doesn’t allow it…hmm… She is not able to walk but from inside she thinks that she prepare meal, do the chores. (C)*



*Not stable, it is increasing. Because, earlier he use to help in house a lot, like putting bins outside and getting it in…he takes lots of time in wearing shoes, will wear it wrong. When we ask him to get shoes off… means he is getting too slow, thinks about it. This is why we don’t allow him to put bins now in a fear that he will fall. (C)*


Subtheme 3.3: Feelings of loneliness

Although most participants with dementia were living with their family, they reported that they felt lonely at times. Having a partner or family member around gave them reassurance in case they forgot something. One caregiver mentioned that her husband was not supportive when she wanted to learn to drive. The husband (person with dementia) said he was fearful of feeling lonely if she went out on her own and left him behind at home.


*If she learn how to drive a car then it will get difficult for me…I will be alone…how will I spend my time? (P)*


Subtheme 3.4: Social stigma and negativity

Most of the participants did not feel part of the group at the community centres they attended and did not wish to talk much about their condition when they were there. Their caregivers related this to feeling negative or being self-consciousness about their health condition.


*She has a little negativity…hmm…she thinks like the other person is thinking bad about her…hmm…it’s her thinking, and feeling…hmm. So she don’t like that kind… of people there. (C)*


Subtheme 3.5: Acceptance and coping strategies

Some participants expressed a sense of acceptance about the fact that their health condition was gradually changing, both physically and mentally. They talked about working out plans to live with it, including developing coping strategies such as reading religious books, giving themselves reminders by making notes, and being happy with their family.


*I have memory problems…just sometimes I forget…a date, where I put something, that’s all, nothing major. (P)*



*I noticed…like…recently I…we read Ramayan*…read Hindi God books…hmm…yes…so it happens sometimes that I have read till here, and why have I started reading that again? …so I do not take stress of this. (P)*


#### 3.1.4. Theme 4: Experience of Support Services

Subtheme 4.1: Satisfaction with home support services

When asked about their home support services, most caregivers were satisfied with the help they received. Services had been arranged depending on the stage of dementia. For example, a person with dementia whose activities of daily living and personal care were restricted received daily home help, assisting them with showering, eating, and dressing.


*First we take the service for 4 days, she goes to <support service> on Friday, hmm… but then she don’t want to go, hmm… then we introduced for fifth day then Saturday and Sunday. So now seven days coming. She got a fall in bathroom, hmmm…it’s pretty hard… so that’s why we put seven days for the shower as well. (C)*



*Yes they come at home sometime to give showers…we have asked for three times only. (C)*


For others, services were provided for activities such as help in driving around and shopping as and when needed. The help was recommended and arranged by the participant’s general practitioner. They particularly liked services being provided in their homes, being subsidised by the government, and being able to talk in their local language with the home help carer and service provider.


*It’s enough. We are getting more than enough …okay…because we are getting help from <support service>…they give help for her. (C)*



*There comes three or four things, like you need physio help…need prescription since I am diabetic…so I need nurse…hmm… so they are always ready. Then they have references…like they referred me for my problem…of this…brain. (P)*


However, not all participants were satisfied with the support received. One described approaching several different services but not receiving adequate support. This led to the family paying someone privately for house cleaning and other domestic help:


*He needs proper support every time people come over here they listen to us but no support … so many times he had a fall so many times we have had the same issue. …we seriously need some support I have someone who comes at the house and does cleaning and everything…. (C)*


Subtheme 4.2: Community centres more suited for people with mild dementia

Community centres were generally seen as useful opportunities for the person with dementia to leave the house. One caregiver mentioned that going to a community centre for Indian senior citizens once a week helped her husband vary his routine. However, she was always worried about him as he was not able to eat properly and was getting physically slow in general. For some people, interacting with other people of the same age group was a source of motivation. Day programmes also allowed caregivers or family members to seek assistance if they needed temporary care for the person with dementia. However, the general view was that the community centres were only suitable for people in the early stages of dementia. Participants reported noticing that with declining health, the attitudes of the people around them began to change, which eventually diminished their interest in attending the centre. It was suggested that the activities in the centres should be more inclusive and suited to people at different levels of cognitive abilities.


*Like today I went, there was a man aged 94 came…at <support service>…hmm… yes…so we feel very happy. Hmmm…he came on wheel chair…so we feel very happy, that this man also wants to live…yes…alright…someone who is 20 years older than me. (P)*



*People were cooperative, but as things are changing now, people are getting cut-off. (C)*


Subtheme 4.3: Aged Residential Care (ARC) does not meet cultural needs and expectations; culturally tailored ARC would make them feel at home

People with dementia and caregivers were asked about their experiences and thoughts about moving into ARC. Although the availability of ARC as an optional service was known, much of the feedback suggested there were several concerns. The concept of ARC, in general, does not fit well with the Indian culture and lifestyle. The two main issues identified were language-related communication barriers and the unavailability of Indian, particularly vegetarian, food. Out of all the caregivers interviewed, there was just one where the person living with dementia had recently been admitted to ARC on her General Practitioner’s (GP) recommendation. Others had either experienced these services during visits to a facility or had heard stories within their social circles. People living with dementia and their caregivers mentioned difficulty in communicating with the management due to not being provided with an interpreter at the facility. As the majority of the other nursing home residents spoke English, communication with other residents was difficult. Furthermore, speaking in different Indian-origin languages such as Hindi, Fiji Hindi, Punjabi, and Gujrati added another level of difficulty for older people.


*Yes, the main problem is that if you are from Fiji then you need someone who speaks Fiji Hindi…Language is the main problem and communication. If you find anyone speaks in your language…so, this is my feeling. (C)*



*Two things are important. Food is important (yes) umm…my mother also and we also in fact everybody…whosoever goes there takes food with them or make food for them…umm and then …umm… talking is important. (C)*


A caregiver whose mother had recently been placed in an ARC observed that the residents spent most of their time by themselves or watching television. He suggested that ARC facilities should have activities planned to make the residents more mobile and active. He also expressed that nursing homes should have tailored support and resources for older people with diverse cultural needs to make them feel more at home, such as being able to perform their religious prayers. Another person living with dementia suggested that there should be efforts made to enable mixing and mingling with other major ethnic community groups to extend their circle and learn more.


*Besides I am thinking that there should be a community here where there should be activities like playing carrom (popular board game in India). He would get better in this way, he would talk or play or look around, there was nothing like this what I would wish to. Not like, you go there and sit, then someone would feed you on time, just keep watching tv. You can watch that while staying at home. (C)*


#### 3.1.5. Theme 5: Caregiver Experiences

The life of a caregiver tended to revolve around their relative, with much of their time spent with them. Out of the 10 families interviewed, caregivers of seven families had taken up the responsibility of caring for their relative. This involved a range of activities such as medication supervision, preparing meals, or reminding the person living with dementia to eat, and assisting and/or supervising during showers to ensure they did not fall or injure themselves in some way. Most had to take on these responsibilities by themselves with only a little help from external services or extended family members.

Subtheme 5.1: Acceptance of duties and responsibilities

An aspect that came across clearly with caregivers was a sense of duty and responsibility to their relative and acceptance of their role as a caregiver. They acknowledged that their relative was suffering from an incurable condition, so they learnt to accept it, try not to react, and instead help them live through it. One caregiver mentioned her husband had a habit of constantly collecting things and placing them in a pile. She recognised this behaviour as part of the effect of dementia and therefore learned not to intervene or become upset by it:


*He has collected so many things on its own. Hmm…So, I do not touch these…I will pick them; he will collect again. (C)*


Most of the caregivers mentioned that they did not wish to admit their relative to ARC and would like to continue providing care for them at home as long as they can:


*There are expectations and responsibility at the children’s end, that’s why she is at home. (C)*


Subtheme 5.2: Impact on caregiver’s personal and social life

Many caregivers expressed that the responsibility of caring for a family member living with dementia impacted their own personal and social life. Some caregivers had quit their jobs, and some were doing a lot of travelling to care for their loved ones, which resulted in strained personal relationships. A caregiver mentioned that their social circle has been restricted:


*So we used to be party people but we slowly slowly restrict our social circle. (C)*


Another caregiver said that their time was mostly spent at home looking after his wife; he mentioned sometimes getting restless staying inside the house and trying his best to go outside to feel better:


*I get bored staying at home…going for a walk around, come back having a look around of the playground. (C)*


Subtheme 5.3: Worries about own health

Caregivers who were older females (spouses) expressed concern about the impact on their health and wellbeing. Older caregivers had their own health problems to deal with and were not receiving support for themselves:


*Now that I have aged, I said I cannot work too much now. (C)*



*I also get tired beta (*child), I am 77, will be 78 in December. So, it’s not all possible for me doing as well. I have got severe arthritis. (C)*



*Have to remind again…that she is human as well, look at her…she is sick as well…hmm… I am also doing (work). (C)*


Some caregivers mentioned a positive aspect of their role as they learned new ways to keep helping their relative and have family support when needed. For example, one caregiver had learnt to drive in order to take her husband for short walks and to shop. Another caregiver mentioned that taking care of her husband helped keep her active in her old age.

## 4. Discussion

Our study provided new insights into the understanding and experience of dementia from the perspective of the Indian community living in Aotearoa, NZ. The main themes were the understanding and conceptualisation of dementia; experience of diagnostic processes; impact on personal and social life (including loneliness, stigma, and community responses); experience of post-diagnostic support services; and the impact on the caregiver in terms of filial duty, stress, and their own well-being.

### 4.1. Understanding of Dementia

Our finding that dementia is commonly thought of as a part of normal ageing has also been described in other studies which included Indian populations living as ethnic minority groups in Western countries [8,9]. Similarly, medical illnesses and external forces as causes of dementia have also been described elsewhere among South Asian ethnic minorities [10,11]. A recent systematic review of understandings of dementia among indigenous people in low- and middle-income countries [12] highlighted that the early stage of dementia was rarely conceptualised as a biomedical condition but was instead seen as cognitive decline that is part of normal ageing, possibly associated with physical or psychological weakness which required traditional caregiving practices. However, if more challenging behavioural and psychological symptoms of dementia emerged then the community responses to dementia were influenced by the stigma associated with mental illness, which could sometimes result in feelings of shame in caregivers [13,14]. Knowledge about dementia might be expected to be better in high-income countries, but another recent systematic review reported that understanding was poor amongst ethnic minority groups, including Indians, in a number of countries, including the UK and USA [15]. This suggests that promoting health literacy around dementia in all countries is an important first step in improving dementia care.

Our study also identified several understandings of dementia that are possibly unique to the Indian culture. The notion of “karma” was spoken of as a way of accepting one’s fate due to their past deeds. This is a commonly held belief in the Indian culture, particularly those practicing the Hindu religion [16]. This belief appeared to allow some of the people in our study a greater sense of acceptance and peace with the type of life they were living.

### 4.2. Experience of the Diagnostic Process

Most participants experienced difficulties and delays in getting a clear diagnosis and/or understanding what the diagnosis meant. Reasons ranged from the reluctance of GPs to screen or refer patients to the long waiting times for scans or the high cost of private scans and tests. Caregivers often had to “push” for a diagnosis, as symptoms were dismissed as a normal part of ageing even by doctors. Delays in obtaining a diagnosis of dementia in ethnic minority groups have also been described elsewhere, with health professionals needing to be prompted and chased up by a family member or carer [9]. Excluding carers from medical appointments and communication meant further delays as people with dementia often forgot the details relayed.

Pathways to care are influenced by beliefs around illness and ageing, low awareness and knowledge of dementia, and stigma, preventing help-seeking for dementia care [9]. A recent qualitative study in Denmark found that ethnic minority populations (including Indian) did not access dementia care services due to poor awareness, stigma, and lack of culturally appropriate services [17]. However, clinical staff are also reported to require more education around dementia [18] as they are often reluctant to give a diagnosis [19] due to the perception that there is nothing more that can be offered.

### 4.3. Experience of Post-Diagnostic Support Services

In our study, the experience of post-diagnostic support services was predominantly positive in that carers were largely satisfied with the provision and level of home-based support services, tying in with their wish for their family member with dementia to remain at home, rather than be transferred to a care home facility. The provision of home support services such as help with showering, shopping, and cleaning was perceived as particularly beneficial by caregivers, allowing them some respite. Our findings differ from a study of South Asian (Indian and Pakistani origin) people living with dementia in Scotland that found overwhelmingly negative responses, reporting a severe lack of support and very little knowledge of dementia and how to manage it [20]. Carers in our study expressed concerns about aged care facilities not being able to cater to the dietary needs of Indian people with dementia and are unable to communicate in the person’s language, leaving them socially isolated. Studies in Indian populations elsewhere have also reported language to be a major barrier in seeking and receiving dementia health services [14,18].

### 4.4. Impact on Personal and Social Life of the Person Living with Dementia

Dementia has a significant impact of the personal and social lives of people living with dementia. Some people experience a loss of independence and fear of loneliness and stigmatisation. Caregivers often reported that, in order to protect the person living with dementia, the family might withdraw from social activities due to the feeling of “exclusion” and stigma they experienced and a perceived lack of understanding within their community. The concept of the stigma associated with dementia is expressed by people across many countries and cultures and often prevents or delays seeking care and secludes families from their wider circles [21,22].

### 4.5. Caregiver Experiences

Caregivers expressed a range of experiences and emotions. Younger caregivers appeared to cope better, even if they had made sacrifices such as giving up their employment and social life. Older caregivers (usually the female spouse) struggled both with the mental and emotional impact of observing their spouse declining in their mental capacity as well as their own health issues. Caregivers of parents with dementia accepted their role as part of their filial duty, a common concept in the Indian and South Asian society [23]. In Asian cultures families are expected to provide care for the person with dementia, either out of affection or out of duty, and to meet cultural needs that are sometimes beyond their capabilities of standard care services [4,24,25]. The concept of extended families is often seen as a way of keeping parents involved in the lives of their children and grandchildren, hence allowing a symbiotic relationship that often benefits the whole family [24]. On the other hand, such living arrangements may be a source of stress, particularly to the female caregivers in a family where one member has dementia and is no longer able to assist with childcare. The availability of home care support services was seen as a crucial service in these circumstances.

### 4.6. Strengths

This study had several strengths. This study captured insights from people with mild to moderate dementia rather than only the caregivers’ perspectives. The interviews were contextually rich as the interviewers were bilingual and were able to grasp the cultural context of the conversations. The interviewers were also aware of the preferred cultural protocols when addressing and speaking with community elders, such as imparting a sense of being respected as an elder during the interview process. The interviews were conducted in the participants’ homes, creating a safe cultural space and allowing participants to speak freely about their experiences. We followed the Consolidated criteria for reporting qualitative research (COREQ) guidelines [26] in the conduct of our study and reporting of our findings.

### 4.7. Limitations

There were some limitations in this study. First, the study was limited to the Auckland region with participants being identified through an Auckland-based memory service and two non-profit community organisations. Hence, the findings may not be generalizable to the whole of Aotearoa, NZ, particularly in relation to health and social services. Second, Indians speak multiple languages and their culture varies among different regions of India and other parts of the world including Fiji. These data may not have captured the whole range of experiences from the Indian diaspora, and language/culture-specific complexities may have been lost.

### 4.8. Implications for Future Research and Service Development

Our study findings were similar to those found in a recent report [27], which found that people with dementia and their caregivers of mostly European background living in Aotearoa, NZ also experienced challenges such as limited knowledge and understanding of dementia, societal stigma, and caregiver strain. Our findings showed that the Indian population had the additional challenges of a lack of culturally appropriate services for Indian people living with dementia and for their caregivers. These areas are addressed in the New Zealand Dementia Mate Wareware Action Plan 2020 to 2025 [28]. The Plan has four main themes: (1) reducing the incidence of dementia; (2) supporting people living with dementia and their caregivers to live their best possible lives, including access to timely diagnosis and support; (3) building accepting and understanding communities including addressing stigma and increasing awareness of dementia; (4) strengthening leadership and capability across the sector. Understanding the conceptualisation of dementia and dementia care in different cultures is an important starting point to enable culturally appropriate policy and service development in this area. The themes identified in this study provide support for implementing the actions to improve dementia care for the New Zealand Indian community.

## 5. Conclusions

In conclusion, this study identified a range of factors that impacted the understanding and lived experiences of dementia in NZ Indian families. Dementia as ‘forgetfulness’ was conceptualised as a normal part of ageing rather than a disease, which led to a level of acceptance. However, when more challenging symptoms of dementia emerged, stigma and responses from the community became a concern. People living with dementia expressed feelings of worry, loneliness, and a loss of independence. Caregiving was regarded as filial duty but older caregivers were also concerned about their own health needs. Getting a diagnosis was difficult, and culturally tailored dementia health and support services were regarded as important unmet needs for this community. These insights highlight the importance of co-creating care services with families living with dementia to maximise their benefit for different ethnic groups.

## Figures and Tables

**Figure 1 ijerph-19-01432-f001:**
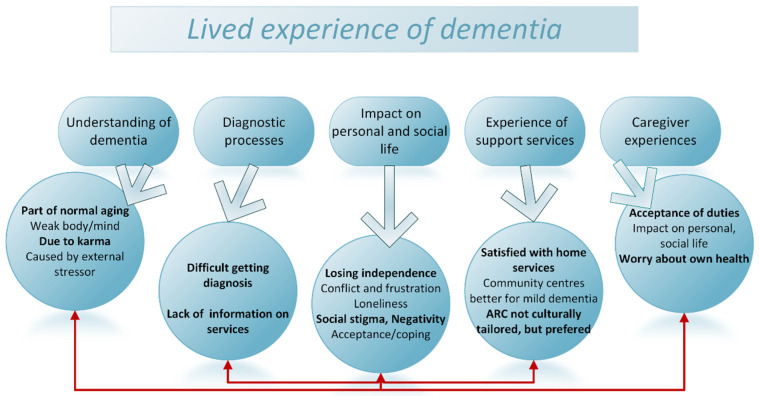
Summary of identified themes and subthemes. ARC: aged residential care.

**Table 1 ijerph-19-01432-t001:** Demographic details of the person living with dementia and caregivers.

Parameters	Person Living with Dementia (*n* = 5)	Caregivers (*n* = 10)
Age, median (range)	74 (65–77)	61 (41–81)
Female, *n* (%)	1 (20%)	6 (60%)
Marital status, *n* (%)		
Married	4 (80%)	9 (90%)
Widowed	1 (20%)	-
Divorced	-	1 (10%)
Birthplace, *n* (%)		
India	2 (40%)	7 (70%)
Fiji	3 (60%)	3 (30%)
Number of years living in NZ, median (range)	11 (10–25)	15 (3–25)
Preferred language, *n* (%)		
Hindi	2 (40%)	5 (50%)
Fiji Hindi	3 (60%)	3 (30%)
English	-	2 (20%)

**Table 2 ijerph-19-01432-t002:** Characteristics of the 10 caregiver participants and their relationship to the person living with dementia.

Caregivers (C)	Characteristics and Relationship to Caregiver
C1	Indian, female, living with son’s family, husband had moderate dementia (P1)
C2	Indian, male, living with his family, mother had severe dementia
C3	Indian, female, living with her son’s family, husband had severe dementia
C4	Indian, female, living with her family, mother had severe dementia
C5	Indian, female, living with her son’s family, husband had mild dementia (P5)
C6	Indian, female, living with daughter’s family, husband had severe dementia
C7	Indian, male, living alone, mother had severe dementia
C8	Fiji Indian, female, living with her family, husband had mild dementia (P8)
C9	Fiji Indian, male, living with his family, father had mild dementia (P9)
C10	Fiji Indian, male, living with his family, wife had mild dementia (P10)

## Data Availability

Not applicable.

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
