# Peer review of "Lived Experience of Dementia in the New Zealand Indian Community: A Qualitative Study with Family Care Givers and People Living with Dementia"

_ijerph, 2022, doi:10.3390/ijerph19031432_

Round 1
Reviewer 1 Report
This qualitative research is about the New Zealand Indian community, how they cope with dementia, and the services that can provide dementia help. The results and considerations that emerged from a minority group are of general interest, because each country is experiencing a growing presence of different ethnic groups. The authors, applying the qualitative approach, were able to highlight the influence of cultural background both on the way of considering dementia and on the accessibility of services, and on the role of the family. They did so, interviewing not only caregivers but also fewer people with dementia, thereby increasing the reader’s interest in the reported results. The authors clearly reported the salient points of the interviews and how they were taken up by the recorded texts. The conclusions are well supported by the data. The article is well written and the subtitles help the reader better understand the results and subsequent discussion.
I have some comments:
1 – Title : generally, it would be preferable to indicate the type of study in the title, qualitative in this case.E.g : Zimmermann BM, Eichinger J, Schönweitz F, Buyx A. Face mask uptake in the absence of mandates during the COVID-19 pandemic: a qualitative interview study with Swiss residents. BMC Public Health. 2021 Nov 26;21(1):2171. doi: 10.1186/s12889-021-12215-4. PMID: 34836517.
2 An "Equator" subgroup produced "Consolidated Criteria for Qualitative Research Reporting (COREQ)" https://www.equator-network.org/reporting-guidelines/coreq/ , which I suggest mentioning in the article, whether you want to follow them or have reasons not to follow these guidelines. (Tong A, Sainsbury P, Craig J. Consolidated criteria for reporting qualitative research (COREQ): a 32-item checklist for interviews and focus groups. Int J Qual Health Care. 2007 Dec;19(6):349-57. doi: 10.1093/intqhc/mzm042. Epub 2007 Sep 14. PMID: 17872937)
3 In the Introduction or discussion, it will be worth explaining a little more about the social status of Indians in New Zealand: do they have a lower income than other ethnic groups? Are the years of education similar to those of other people in New Zealand? Does the housing reflect that of the general population?
These answers can be useful in understanding whether low access to services is explained only by cultural background or also by some form of social exclusion, beyond language.
Author Response
Reviewer 1:
This qualitative research is about the New Zealand Indian community, how they cope with dementia, and the services that can provide dementia help. The results and considerations that emerged from a minority group are of general interest, because each country is experiencing a growing presence of different ethnic groups. The authors, applying the qualitative approach, were able to highlight the influence of cultural background both on the way of considering dementia and on the accessibility of services, and on the role of the family. They did so, interviewing not only caregivers but also fewer people with dementia, thereby increasing the reader’s interest in the reported results. The authors clearly reported the salient points of the interviews and how they were taken up by the recorded texts. The conclusions are well supported by the data. The article is well written and the subtitles help the reader better understand the results and subsequent discussion.
Thank you for your kind comments
I have some comments:
1 – Title : generally, it would be preferable to indicate the type of study in the title, qualitative in this case.E.g : Zimmermann BM, Eichinger J, Schönweitz F, Buyx A. Face mask uptake in the absence of mandates during the COVID-19 pandemic: a qualitative interview study with Swiss residents. BMC Public Health. 2021 Nov 26;21(1):2171. doi: 10.1186/s12889-021-12215-4. PMID: 34836517.
We have amended the title to read: “Lived experience of dementia in the New Zealand Indian Community: a qualitative study with family care givers and people living with dementia”
2 An "Equator" subgroup produced "Consolidated Criteria for Qualitative Research Reporting (COREQ)" https://www.equator-network.org/reporting-guidelines/coreq/ , which I suggest mentioning in the article, whether you want to follow them or have reasons not to follow these guidelines. (Tong A, Sainsbury P, Craig J. Consolidated criteria for reporting qualitative research (COREQ): a 32-item checklist for interviews and focus groups. Int J Qual Health Care. 2007 Dec;19(6):349-57. doi: 10.1093/intqhc/mzm042. Epub 2007 Sep 14. PMID: 17872937)
We have added that “We followed the COREQ guidelines [26] in the conduct of our study and reporting of our findings.” Lines 489-90, page 13.
We also added more details about the study as follows:
“both male and female” (lines 85-86, page 2)
“The researchers introduced themselves to the participants and provided a brief background about themselves prior to the interview, including the reasons for their own interest in the area.” (lines 99-101, page 3)
“In all interviews only the interviewers and interviewees were present, but in one interview, other family members were present (daughter and grandchildren) however they did not participate.” (lines 133-135, page 3)
“We have not included the participant IDs with quotations in the report as it would have been easy to re-identify participants from the sociodemographic tables due to the small population living in the community.”(lines 154-156, page 4)
3 - In the Introduction or discussion, it will be worth explaining a little more about the social status of Indians in New Zealand: do they have a lower income than other ethnic groups? Are the years of education similar to those of other people in New Zealand? Does the housing reflect that of the general population?
These answers can be useful in understanding whether low access to services is explained only by cultural background or also by some form of social exclusion, beyond language.
We have added the following section in lines 37-45 of page 1 and 2. “The majority of Indians (approx. 75%) were born overseas. Indians in NZ are proportionally much more likely to be younger, and to have more post-high school qualifications than the national average. Approximately 60% are employed full-time, however, their mean income is lower than the national average. In terms of origin, the majority of Indians in NZ hail from India, while a smaller but substantial proportion are from Fiji (described as Fijian Indian). Fijian-Indians are an ethnic group from the Pacific Islands of Fiji. (Ref Stats NZ https://www.stats.govt.nz/tools/2018-census-ethnic-group-summaries/fijian-indian).”
Reviewer 2 Report
General comments:
I believe that the problem of the article is relevant and current in nursing and in the general population due to the high incidence of dementia and the expected data in the coming years
Title
The title reflects the content and problem studied.
Abstract
The abstract meets the standards of the Journal. Reference is made to the methodology used (that is, the research carried out, where the sample was recruited, interview methods, etc.), the main results and conclusions. The implications of the research are addressed.Key Words
Keywords
The keywords are representative of the subject studied and exposed
Introduction
A state of the art of dementia is made in relation to the study. The objective of the study is mentioned, as well as the justification for the choice and importance of studying this theme
Methods
There is a detailed description of the research methods used. The design is correct. The authors show the participants and settings, how the interviews (semi-structured) were, as well as the data analysis and approval by the ethics committee.
Results
The results shown are concrete and detailed. They show tables with sociodemographic data as well as statistics for both dementia and caregivers. They also show the topics analyzed and relate them to dementia. They describe the interviews in detail. Well described results.
Discussion
the discussion is extensive and reasoned
Conclusions
The conclusions synthesize the results obtained regarding dementia in patients and caregivers in a clear and convincing way.
Limitations and Implications for future research and service development
The authors show the limitations and implications for future research of the study in patients with dementia and their caregivers.
References
The references used are current, the vast majority dating from the last five to ten years.
Author Response
No comments to address. We thank you for your kind feedback
Reviewer 3 Report
Thank you for inviting me to review the manuscript entitled: "The understanding and lived experiences of dementia in the New Zealand Indian Community". It is a manuscript that addresses a topic that is very important today and predictably even more relevant in several years due to the aging of the population. The following are a series of considerations, observations, and recommendations:
TITLE
A revision is recommended. It is not really long, its length is adequate and it provides sufficient information. However, its translation into English and the typical style of scientific article writing make its revision relevant, since it is not usual to begin an article title with "the".
SUMMARY
Sociodemographic information on the number of people, mean age and standard deviation, percentage of women or men, etc. is missing.
KEYWORDS
If all words are of equal importance, the choice should be made to put them in alphabetical order.
INTRODUCTION
The information provided is relevant but insufficient. A much more in-depth approach to the topic is needed, analyzing the nature of the topic, explanatory theories, etiology, related variables that have also been analyzed in this study, conceptualization, typical measurement instruments in other research that are related to the present study, etc. In addition, it is too brief an introduction.
MATERIALS AND METHODS
Sociodemographic information is missing.
Inclusion criteria appear but it would be necessary to include exclusion criteria as well. The sociodemographic variables in Table 1 of the Results section should appear in the description of the sample.
It would be interesting to further describe the software used (Computer Assisted Qualitative Data Analysis Software, NVivo (Version 12, QSR International) for coding) and even to justify in the introduction its use with other previous studies. Likewise, the appropriate subsection to do so would be that of data analysis.
The instrument is not clear, the questions do not appear, nor does the reliability or internal consistency of the tool.
RESULTS
A study with only 10 subjects is very difficult to generalize.
DISCUSSION
Although the results are well discussed, they may be insufficient to understand the underlying problem.
The authors' hard work in this section in structuring the information, describing in detail the strengths, limitations, etc., should be highlighted.
CONCLUSIONS
It is consistent with the rest of the study.
In summary, this is an interesting research but needs to be further elaborated in several aspects. Thank you for your attention.
Author Response
Thank you for inviting me to review the manuscript entitled: "The understanding and lived experiences of dementia in the New Zealand Indian Community". It is a manuscript that addresses a topic that is very important today and predictably even more relevant in several years due to the aging of the population. The following are a series of considerations, observations, and recommendations:
TITLE
A revision is recommended. It is not really long, its length is adequate and it provides sufficient information. However, its translation into English and the typical style of scientific article writing make its revision relevant, since it is not usual to begin an article title with "the".
We have amended the title to read: “Lived experience of dementia in the New Zealand Indian Community: a qualitative study with family care givers and people living with dementia”
SUMMARY
Sociodemographic information on the number of people, mean age and standard deviation, percentage of women or men, etc. is missing.
We have replaced the sentence in the abstract "Participants and caregivers were recruited from a hospital memory service, and two not-for-profit community organisations in Auckland, Aotearoa NZ" with “Ten caregivers (age range: 41-81) and five people living with mild dementia (age range: 65-77) were recruited from a hospital memory service, and two not-for-profit community organisations in Auckland, Aotearoa NZ” (page 1, lines 16-18)
KEYWORDS
If all words are of equal importance, the choice should be made to put them in alphabetical order. We have changed the key words to "caregivers; dementia; Indian community; New Zealand; qualitative"
INTRODUCTION
The information provided is relevant but insufficient. A much more in-depth approach to the topic is needed, analyzing the nature of the topic, explanatory theories, etiology, related variables that have also been analyzed in this study, conceptualization, typical measurement instruments in other research that are related to the present study, etc. In addition, it is too brief an introduction.
We respectfully request more specific information relating to "explanatory theories, etiology, related variables that have also been analyzed in this study, conceptualization, typical measurement instruments" For example, this is a qualitative study about the lived experience of dementia in a specific population and therefore does not use measurement instruments.
MATERIALS AND METHODS
Sociodemographic information is missing. The sociodemographic information is in tables 1 and 2, and also in the first two paragraphs of the results section.
Inclusion criteria appear but it would be necessary to include exclusion criteria as well. We have added that "We aimed to be as inclusive as possible but we excluded people who were residing in care homes, or were unable to be interviewed in English, Hindi, or Fijian Hindi." (page 2, lines 80-82) in addition to the pre-existing lines 82-84.
The sociodemographic variables in Table 1 of the Results section should appear in the description of the sample. The sociodemographic variables in Table 1 of the Results section do appear in the description of the sample in the text of Results section.
It would be interesting to further describe the software used (Computer Assisted Qualitative Data Analysis Software, NVivo (Version 12, QSR International) for coding) and even to justify in the introduction its use with other previous studies. Likewise, the appropriate subsection to do so would be that of data analysis. NVivo is a tool to organise qualitative data and therefore does not require justification for its use in previous studies. We have added clarification on line 110 on page 3: “The transcripts were uploaded into a Computer Assisted Qualitative Data Analysis Software, NVivo (Version 12, QSR International) for organisation of the data for coding [6].
The instrument is not clear, the questions do not appear, nor does the reliability or internal consistency of the tool. There is no instrument – this is a qualitative study. We did include a topic guide, but this was just a guide for interviewers. To clarify we have changed the text on page 2 line 95 to say “The Topic guide (Appendix A) guided interviewers' explorations of the understanding of dementia, experience of the diagnostic process, adjustments made to lifestyle, available support services, and any concerns or recommendations. Caregivers were also asked about their caregiving experiences, any concerns, or issues they were facing and, satisfaction with the level of formal and informal support received.”
RESULTS
A study with only 10 subjects is very difficult to generalize. The study had 15 subjects which is usually considered sufficient to reach data saturation in qualitative studies. We have added "This was sufficient to reach data saturation." on line 135 on page 3 to clarify this.
DISCUSSION
Although the results are well discussed, they may be insufficient to understand the underlying problem.
The authors' hard work in this section in structuring the information, describing in detail the strengths, limitations, etc., should be highlighted.
Thank you for your kind comments
CONCLUSIONS
It is consistent with the rest of the study.
In summary, this is an interesting research but needs to be further elaborated in several aspects. Thank you for your attention.
Thank you for your kind comments
Round 2
Reviewer 3 Report
Thank you very much for your efforts to improve various aspects of the manuscript. However, the aspects mentioned still need to be improved. In summary, the manuscript needs more theoretical background, not only in terms of length, but also in the number of bibliographic references used. This influences above all the introduction but also the discussion of the data. On the other hand, I consider it pertinent to further justify the small number of participants. Thank you for your attention.
Author Response
Thank you very much for your efforts to improve various aspects of the manuscript. However, the aspects mentioned still need to be improved. In summary, the manuscript needs more theoretical background, not only in terms of length, but also in the number of bibliographic references used. This influences above all the introduction but also the discussion of the data. On the other hand, I consider it pertinent to further justify the small number of participants.
Thank you for your comments.
We feel there is sufficient information and references in the discussion for the reader to understand our methodological approach in this qualitative study, and to understand our discussion of the findings. We recruited 15 participants in the study which is generally considered sufficient to reach data saturation.